# ATTRIBUTE RECOGNITION WITH IMAGE-CONDITIONED PREFIX LANGUAGE MODELING

## ABSTRACT

Predicting object identity and visual attributes is a fundamental task in many computer vision applications. While large vision-language models such as CLIP had largely solved the task of zero-shot object recognition, zero-shot visual attribute recognition remains challenging because CLIP's contrastively learned language-vision representation does not effectively encode object-attribute dependencies. In this paper, we revisit the problem of attribute recognition and propose a solution using generative prompting, which reformulates attribute recognition as the measurement of the probability of generating a prompt expressing the attribute relation. Unlike contrastive prompting, generative prompting is order-sensitive and designed specifically for downstream object-attribute decomposition. We demonstrate through experiments that generative prompting consistently outperforms contrastive prompting on two visual reasoning datasets, Visual Attribute in the Wild (VAW) and a proposed modified formulation of Visual Genome, which we call Visual Genome Attribute Ranking (VGAR).

## 1 INTRODUCTION

Understanding attributes associated with objects in an image is essential for many computer vision applications, including content recommendation, visual reasoning, and text-to-image models. While supervised learning techniques such as classification (Simonyan & Zisserman, 2015; Krizhevsky et al., 2017; He et al., 2016), detection (Ren et al., 2015; Redmon et al., 2016; He et al., 2017), and segmentation models (Ronneberger et al., 2015; Chen et al., 2017) have made significant progress in object recognition tasks, directly adding a branch for object-agnositic attribute prediction (Farhadi et al., 2009; Ferrari & Zisserman, 2007; Patterson & Hays, 2016) can result in irrelevant or counterfactual outputs since it fails to model the inter-dependency between attributes and objects. Other existing attribute learning methods rely heavily on human annotations (Zhang et al., 2021; Anderson et al., 2018; Lampert et al., 2009; Jayaraman & Grauman, 2014; Al-Halah et al., 2016) to address this dependency, yet this makes them expensive and hard to scale. In summary, the problem of establishing object-attribute relationships at scale remains unsolved.

Large-scale image-text foundation models such as CLIP (Radford et al., 2021) and ALIGN (Jia et al., 2021) inspired us to explore their potential for attribute learning. These models have learned from vast amounts of noisy image-text pairs from the web, adequately utilizing self-supervised learning to benefit from easily accessible data sources. They have shown exceptional performance in zero-shot object recognition (Rao et al., 2022; Yao et al., 2021; Zhong et al., 2022; Wang et al., 2022; Ma et al., 2022; Shi et al., 2022; Materzyńska et al., 2022; Pham et al., 2022) through image-text similarity measurement, a method which we refer to as "contrastive prompting".

However, naively applying contrastive prompting to attribute prediction tasks yields suboptimal performance due to its two inherent problems. First, treating input text as a unstructured whole results in incomplete representations to be learned. Since the model is only trained to match image-text pairs, it tends to overlook visual attributes if the object in the text is distinguishable enough in the image. This creates a discrepancy between the pre-training and the downstream tasks: the model learned to primarily differentiate between objects but is later asked to understand finer attributes. Second, contrastive prompting cannot model the co-dependency between objects and attributes. Since contrastive pre-training does not capture word sequence order, as opposed to language model pre-training (Fig. 1 (left)), the model is unable to filter out counterfactual combinations such as "bell-

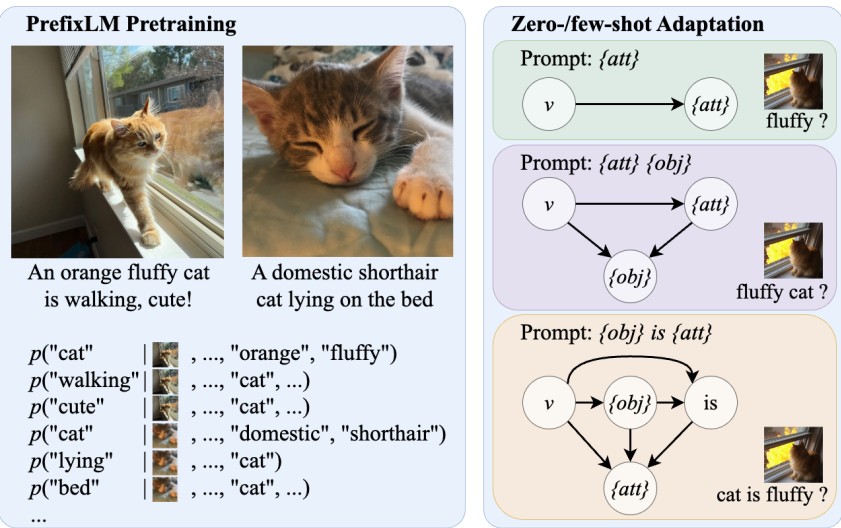

Figure 1: Prefix language modeling and generative prompting. During pre-training, the image-conditioned prefix language model (prefixLM) learns to caption images, and curates knowledge regarding object-attribute compositions. In the downstream attribute recognition, a novel generative prompting strategy is used to rearrange knowledge and to extract reasoning results from prefixLM. Different from the contrastive prompting, generative prompting models the conditional dependence, hence is more representative of factual knowledge.

shaped sky" or "sky (for) parking" (see Fig. 4) and still produces high-confidence scores. These challenges emphasize the necessity for better methods in modeling object-attribute dependency in the context of image-text foundation models.

This paper presents a novel approach to address the two aforementioned problems in applying image-text foundation models to attribute learning. The approach consists of two parts: prefix language modeling (prefixLM) (Bengio et al., 2000; Wang et al., 2021) as the pre-training foundation and a novel text generation-based prompting formulation for extracting structural reasoning information (see Fig. 1). During pre-training, the prefixLM is trained to predict the next token based on visual inputs and previous texts, which inherently captures diverse combinations of object-attribute dependencies. In the downstream attribute recognition task, we measure the object-attribute alignment in an image by evaluating the probability of generating the relational prompts. We refer to this approach as "generative prompting". In particular, this method enables flexible prompting for a wide range of attribute relations (associative, possessive, or further modified with temporal words like "currently" or "already") through building arbitrary conditional dependency models (Fig. 1 (right)) for downstream tasks at inference time, which are effectively "meta-models".

We formulate two immediate applications for the proposed prefixLM + generative prompting framework: (1) describing objects through their visual appearance, state of being, or relationship to other objects in the image. And conversely, (2) recognizing objects based on their various visual attributes such as color, shape, size, and so on. In addition, our method can be further applied towards many other visual tasks that require structural reasoning. We summarize the contributions as follow:

- We demonstrate the limitations of contrastive learning in capturing object-attribute dependencies.
- We establish the effectiveness of using prefixLM as a foundational model for capturing complex object-attribute relationships in pre-training.
- We propose a novel generative prompting mechanism that explicitly models the dependencies between objects and attributes, We show that the generative prompting can serve as a meta-model for attribute recognition to create different probabilistic models.
- We introduce Visual Genome Attribute Ranking (VGAR), a novel benchmark encompassing both attribute and object recognition tasks in a unified setting, to demonstrate the generalizability of the proposed approach.

## 2 RELATED WORK

We first introduce studies on attribute learning, which mostly rely on hand-crafted probabilistic models without the use of large language models. Then, we summarize the language modeling works to specifically introduce PrefixLM to the attribute learning tasks. Finally, we provide an overview of existing prompt learning techniques to introduce our novel approach of generative prompting, which distillate information from the pretrained PrefixLM.

**Visual attribute recognition** involves identifying the properties of visual objects, such as their color, material or shape. Early works had focused on object description ($img{\rightarrow}att$) using classification (Farhadi et al., 2009; Ferrari & Zisserman, 2007; Patterson & Hays, 2016) or relative ranking models (Parikh & Grauman, 2011; Kovashka et al., 2012; Chen & Grauman, 2018; Wang et al., 2016; Yu & Grauman, 2014) to learn the attributes strength independent of object category. Some works have used attributes as a foundation (Lampert et al., 2009; Jayaraman & Grauman, 2014; Al-Halah et al., 2016) for zero-shot object recognition ($img,att{\rightarrow}obj$; e.g., recognizing "zebra" by the attributes "black", "white", "striped"). They learned an attribute vector space and used it to recognize unseen visual objects based on the marginal probability. In visual object detection, models (Zhang et al., 2021; Anderson et al., 2018) have trained attribute prediction branches using the Viusal Genome dataset (Krishna et al., 2017) to improve model diversity and to create models with multi-tasking capabilities. These models concatenate the visual feature with the ground-truth object class embedding and feed them into an attribute prediction branch ($img,obj{\rightarrow}$att).

Vector space-based approaches had also been studied for attribute recognition. For example, Pham et al. (2022); Gu et al. (2021); Chen et al. (2023a) applied the CLIP (Radford et al., 2021). They use the CLIP embedding to compare visual information to predefined attribute prompts ($img{\leftrightarrow}obj,att$), to determine if the image contains those attributes. In addition to CLIP, Nagarajan & Grauman (2018); Naeem et al. (2021); Misra et al. (2017); Nan et al. (2019) allow objects and attributes to be projected into the same feature space, while the decoration of attributes on objects is modeled as an operator ($img{\leftrightarrow}obj$ OP $att$, operator OP could be $\pm$ or linear transform).

Our innovation lies in the novel view of treating attribute recognition as a language modeling problem. We integrate probability modeling for image, object class, and attribute prediction in an unified image-text model, while leveraging LLM's foundational pre-training.

**Language modeling** (LM) estimates the probability of a sequence of words being observed in a sentence. Early language models used dense neural networks (Bengio et al., 2000) or recursive neural networks (Xu et al., 2015), while recent large language models (LLMs) (Devlin et al., 2019; Yu et al., 2022; Wang et al., 2021; Radford et al., 2021; Chowdhery et al., 2022) are based on the transformer architecture (Vaswani et al., 2017) because of its strong generalizability. LM has many applications in both NLP and computer vision, including question answering(QA) (Rajpurkar et al., 2016; Yang et al., 2015), conversational question answering (Reddy et al., 2019), visual captioning (Young et al., 2014; Chen et al., 2015; Sharma et al., 2018; Agrawal et al., 2019), and visual question answering (Antol et al., 2015; Zhang et al., 2016; Goyal et al., 2017). These applications can be categorized into three main types of LM: (1) image-text matching (Frome et al., 2013), (2) masked language modeling (Devlin et al., 2019), and (3) prefix language modeling (Bengio et al., 2000).

Attribute recognition is a condensed VQA problem that requires predicting the visual attribute of an query object. The foundational methods proposed in the VQA domain mostly combine image-text matching and masked language modeling. Examples include LXMERT (Tan & Bansal, 2019), UNITER (Chen et al., 2020), OSCAR (Li et al., 2020), VinVL (Zhang et al., 2021), ViLT (Kim et al., 2021), VLMo (Bao et al., 2022).

Different from these works, we show that prefix language modeling (prefixLM) (Bengio et al., 2000; Yu et al., 2022; Wang et al., 2021; Chen et al., 2022; 2023b)) can approximate masked language modeling (see Sec. 3.3) in the attribute tasks. With a novel prompting scheme, prefixLM exhibits even greater expressive power than MLM, making it a powerful tool for deep visual reasoning (Yao et al., 2021; Tsimpoukelli et al., 2021).

**Prompt learning** originated in the field of natural language processing (NLP), where tasks like question-answering are frequently formulated as a "fill-in-the-blank" problem. Notable examples include BERT (Devlin et al., 2019) which employs masked language modeling, and GPT (Radford

et al., 2019) that uses prefix language modeling. While large language models (LLMs) (Chowdhery et al., 2022; Thoppilan et al., 2022; OpenAI, 2023) have been widely explored in NLP for fact-based reasoning, their application in the computer vision domain is relatively unexplored.

Prompt learning in computer vision has gained attention following the success of CLIP (Radford et al., 2021). Numerous works (Rao et al., 2022; Yao et al., 2021; Zhong et al., 2022; Wang et al., 2022; Ma et al., 2022; Shi et al., 2022; Materzyńska et al., 2022; Pham et al., 2022) have focused on designing CLIP-prompts or utilizing the pre-trained CLIP checkpoint. Approaches such as Zhou et al. (2022a;b) learn the prompting vectors instead of manually designing text prompts.

Our approach focus on its application towards attribute learning, which the aforementioned contrastive learning based methods are ill-suited for. Our proposed generative prompting is based on image-conditioned prefix language modeling (Yu et al., 2022; Wang et al., 2021), which takes sequence ordering into consideration and is therefore well-suited for modeling the dependence between visual objects and attributes. The proposed method has potential applications in other visual reasoning problems such as visual relation detection (Lu et al., 2016) or scene graph generation (Johnson et al., 2015).

## 3 APPROACH

### 3.1 IMAGE-CONDITIONED LANGUAGE MODELING

Our proposed generative prompting is based on image-conditioned prefix language modeling, i.e. image captioning. Given an image $v$, we aim to generate the corresponding text $x = (s_1, ..., s_n)$ by modeling the probability $p(x|v)$ using Eq. 1. This equation factors $p(x|v)$ into the product of conditional probabilities (Bengio et al., 2000; Radford et al., 2019), where at each time step, the model predicts the next token $s_i$ based on the visual input $v$ and previous tokens $(s_0, ..., s_{i-1})$ ($s_0$ is the start-of-sentence token "").

$$p(x|v) = \prod_{i=1}^{n} p(s_i|v, s_1, \ldots, s_{i-1})$$

(1)

The factorization provided by Eq.1 is advantageous as it breaks down the word generation process into individual probability factors. In Fig. 1 (left), we show that the model can capture various object-attribute compositions during pre-training. As a result, in downstream attribute-related tasks, we can leverage this factorization to address reasoning questions such as $p(w_{att}|v, w_{obj})$, which represents the probability of observing an attribute $w_{att}$ (e.g., "orange", "fluffy") given the visual input $v$ and object $w_{obj}$ (e.g., a "cat").

### 3.2 GENERATIVE PROMPTING FOR ATTRIBUTE CLASSIFICATION

We formalize the prompt-based classification task to establish a common foundation for both generative prompting and contrastive prompting. Specifically, given an image $v$ and text prompts $t^{(1)}, \ldots, t^{(C)}$ ($C$ is number of classes), prompt-based classification involves designing a loss function $L(v, t)$ to measure the cost of aligning image $v$ and text $t^{(i)}$ ($1 \leq i \leq C$). Thus, zero-shot classification can be achieved by finding the class label $c = \operatorname{argmin}_{1 \leq i \leq C} \{L(v, t^{(i)})\}$.

**Contrastive prompting** builds on the fact that paired image-text are projected into the same feature space through contrastive learning during pre-training. Assuming the image is encoded as $f(v)$ and the text is encoded as $g(t)$, the contrastive learning objective aims to maximize the inner product between the matched image-text embeddings while minimizing the unmatched ones. This encourages paired image-text samples to have a high similarity while pushing unpaired samples apart. Under the common assumption of unit norm in the embeddings (Sohn, 2016; Jia et al., 2021; Radford et al., 2021), this can be equivalently represented by using the L2 loss to measure the distance between image and text, denoted as $L^{(con)}(v, t) = \|f(v) - g(t)\|_2$.

**Generative prompting** is our proposed approach for visual attribute recognition, which utilizes the cross-entropy to evaluate the image-text alignment loss, represented as $L^{(gen)}(v, t) = -\sum_{i=1}^{N} \hat{p}(t_i) \log q_\theta(v, t_{j|j<i})$. Here, $\hat{p}(t_i) \in \mathbb{R}^{1 \times V}$ ($1 \leq i \leq N$, $N$ is the length) represents the

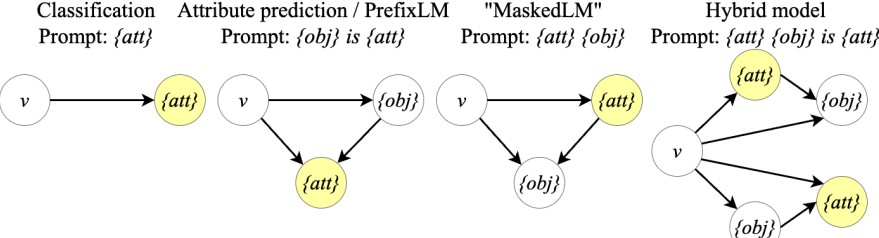

Figure 2: Conditional dependencies that different generative prompts modeled. Attribute recognition is modeled as a fill-in-the-blank problem for the highlighted "{att}" in the graph. Our proposed generative prompting optimizes or approximates the joint probability of observing the graph, while it only relies on the prefixLM pre-training.

one-hot representation of the $i$-th token of prompt $t$. To generate the information at the $i$-th step, the model $q_\theta$ relies on the image $v$ and all previous text tokens $t_{j|j<i}$ to produce a probability distribution $q_\theta(v, t_{j|j<i}) \in \mathbb{R}^{V \times 1}$ over the vocabulary $V$. The term $-\hat{p}(t_i) \log q_\theta(v, t_{j|j<i}) \in \mathbb{R}^1$ represents the cross-entropy between the $i$-th token in the prompt $t$ and the model's prediction at the $i$-th step. Fig. 3 (middle) provides a visual representation of this equation.

## 3.3 Modeling the Conditional Dependence

We can build different probabilistic models for visual attribute recognition by changing the words or word ordering in the prompts (see Fig. 2). Our key contribution to the visual attribute learning community is proposing and showcasing the versatility of the generative prompting, which enables the designs of arbitrary probabilistic model via prompt engineering.

**Prompt "{att}".** This prompt models the simplest dependency for predicting attribute based on the image. In this dependency model, we focus on the cross-entropy of classifying the image as having a specific attribute, which can be achieved through a simple classification model. This approach aligns with early methods (Farhadi et al., 2009; Ferrari & Zisserman, 2007; Patterson & Hays, 2016; Parikh & Grauman, 2011; Kovashka et al., 2012; Chen & Grauman, 2018; Wang et al., 2016; Yu & Grauman, 2014) that describe attributes rather than naming the objects.

**Prompt "{obj} is {att}".** This prompt models the prediction of attributes based on both an image and an object, approximating $p("\{att\}"|v, "\{obj\}")$. In this dependency model, all prompts share the same prefix "{obj} is" (e.g., "cat is orange", "cat is fluffy", "cat is cute", etc.). Therefore, the only factor that matters in generative prompting becomes $-\hat{p}("\{att\}")q_\theta(v, "\{obj\}", "is")$, which quantifies the loss associated with classifying an attribute given the image and object. This dependency model characterizes recent attribute works such as Zhang et al. (2021); Anderson et al. (2018); Pham et al. (2021; 2022).

**Prompt "{att} {obj}".** This prompt is similar to Masked Language Modeling (MLM) (Devlin et al., 2019) as it involves filling in the blank in a sentence like "an image of a [MASK] cat". However, there are two key distinctions: (1) $p("\{att\}"|v)$ requires the attribute must be easily recognizable from the image, and (2) $p("\{obj\}"|v, "\{att\}")$ requires that the attribute can be employed to modify the object. In contrast, MLM uses all contextual information to predict the masked token (attribute), regressing to the earlier generative prompt "{obj} is {att}"). The probabilistic modeling derived from the prompt "{att} {obj}" closely resembles the approaches in Lampert et al. (2009); Jayaraman & Grauman (2014); Al-Halah et al. (2016), where attributes were utilized for object recogntion.

**Prompt "{att} {obj} is {att}".** This prompt produces unconventional sentences such as "fluffy cat is fluffy". We highlight this prompt to showcase the versatility of generative prompting. In essence, it encompasses all three previously discussed conditional probability terms: (1) $p("\{att\}"|v)$ – classification; (2) $p("\{obj\}"|v, "\{att\}")$ – object-attribute compatibility; and (3) $p("\{att\}"|v, "\{obj\}")$ – attribute prediction based on image and object. We present an approximate probability graph in Fig. 2 (right), where we duplicate both the attribute and object nodes. With the duplicated "{att}" in the prompt, the resulting modeling accounts for the co-dependency between object and attribute. For example, attributes preceding objects: "red car", "blue sky"; and objects preceding attributes:

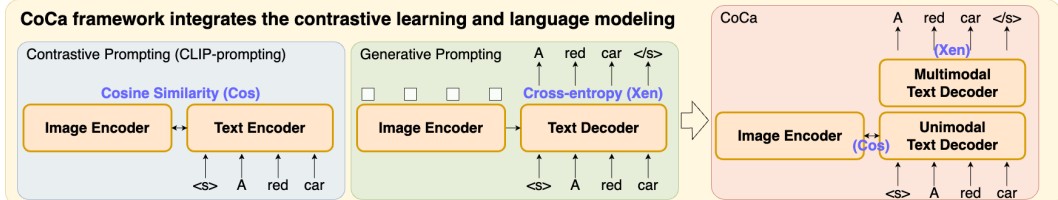

Figure 3: Overview of Coca. The CoCa model integrates both contrastive learning and prefix language modeling. While its text decoder as a whole (Unimodal+Multimodal) learns to caption images, the first few layers (Unimodal) can be used for contrastive learning.

"kid is smiling", "cat is lying". This formulation further bridges the gap between pre-training and zero-shot inference.

**Discussion.** Our proposed generative prompting is novel for two reasons. Firstly, from the language modeling perspective, we offer a new solution for training using prefixLM, enabling the mimicking of MLM or more advanced LM in a zero-shot manner for downstream tasks. Secondly, our generative prompting produce dependency models at inference time that serves as meta-models for attribute recognition, since we can flexibly modify the probabilistic modeling and conditional dependence through changes in text prompts. In our experiments, we show the results for the four different probabilistic attribute models ( Fig. 2).

### 3.4 FINETUNING ON ATTRIBUTE TASKS

Since the attribute class names have similar lengths, their cross-entropy scores $L^{(gen)}(v, t)$ with the image are expected to fall within a similar range of values (see Fig. 4). Therefore, an intuitive way to adapt the knowledge in a few-shot manner is to learn to "rescale" the prompting scores to adapt to the new dataset priors during finetuning. Specifically, if $t^{(c)}$ is the prompt for class $c$, we introduce learnable parameters bias $\mu_c$ and scaling factor $\sigma_c$ to adjust the $L^{(gen)}(v, t^{(c)})$, resulting in a transformed probability $p_c = \text{sigmoid}\left(-\frac{L^{(gen)}(v, t^{(c)}) - \mu_c}{\sigma_c}\right)$. This probability $p_c$ represents the likelihood of the image-object pair being associated with attribute $c$. During finetuning, $p_c$ can be optimized using cross-entropy loss. In Sec. 4.2, we also provide the baseline results of finetuning the contrastive prompting, using the same approach (but with loss score $L^{(con)}$ instead of $L^{(gen)}$).

## 4 EXPERIMENTS

### 4.1 IMPLEMENTATION DETAILS

We use CoCa (Yu et al., 2022) pretrained on the LAION (Schuhmann et al., 2022) as the foundation model. CoCa combines multimodal contrastive learning with image-conditioned prefix language modeling, as illustrated in Fig. 3. Its text decoder consists of (1) a unimodal text decoder trained on a contrastive learning objective with an image encoder, and (2) a multimodal text decoder trained to generate image captions by cross-attending to the image encoder. We adopt CoCa as the foundation model as it allows for both contrastive and generative prompting in one model trained on the same image-text data, ensuring a fair comparison. In our experiments, we use the CoCa Base model, which consists of a ViT (Kolesnikov et al., 2021) image encoder with 12 transformer layers, a unimodal text decoder with 6 layers, and a multimodal text decoder with an additional 6 layers. The image resolution is set to $224 \times 224$ pixels with a patch size of $16 \times 16$ pixels. All transformer layers have hidden dimensions of 768 and MLP size of 3,072. Additional details on modeling and fine-tuning can be found in the supplementary materials.

The following two datasets are used for evaluation:

**Visual Attribute in the Wild (VAW)** (Pham et al., 2021) is a large dataset of images with explicitly labeled positive and negative attributes. The associated task requires a model to predict a set of visual attributes given an object's name and the image it is on. VAW contains 216,790 objects from 58,565 images for training, 12,286 objects from 3,317 images for validation, and 31,819 objects from 10,392 images for testing. We use the test set to evaluate zero-shot attribute recognition.

**Visual Genome Attribute Ranking (VGAR)** is a modified version of the Visual Genome (VG) dataset (Krishna et al., 2017) also designed to evaluate a model's ability to recognize visual at-

Table 1: Zero-shot results on the VAW dataset. **Blue** and **black** numbers in bold represent the best and second best, respectively.

|  | Prompt Templates | Rank↓ | mR$^{@15}$↑ | mAP↑ |
|---|---|---|---|---|
| Cont. | "{att}" | 95.1 | 32.0 | 52.5 |
|  | "{att} {obj}" | 149.8 | 22.4 | 47.1 |
|  | "{obj} is {att}" | 151.4 | 23.2 | 45.9 |
|  | "{att} {obj} is {att}" | 141.0 | 23.7 | 48.3 |
| Gene. | "{att}" | 82.1 | 28.7 | **53.8** |
|  | "{att} {obj}" | 63.9 | **35.9** | 47.7 |
|  | "{obj} is {att}" | **61.9** | **32.9** | 46.1 |
|  | "{att} {obj} is {att}" | **56.0** | 31.7 | **49.9** |

Table 2: Finetuning results on the VAW dataset. The best result by rank across prompting methods and prompt templates is highlighted in gray.

|  | Prompt Templates | Rank↓ | mR$^{@15}$↑ | mAP↑ |
|---|---|---|---|---|
| Cont. | "{att}" | 18.3 | 48.6 | 69.6 |
|  | "{att} {obj}" | 12.8 | 59.8 | 65.7 |
|  | "{obj} is {att}" | 12.3 | 58.9 | 66.7 |
|  | "{att} {obj} is {att}" | 12.2 | 59.6 | 67.3 |
| Gene. | "{att}" | 18.0 | 50.5 | 71.7 |
|  | "{att} {obj}" | 11.4 | 61.8 | 70.8 |
|  | "{obj} is {att}" | **11.1** | 62.1 | **72.0** |
|  | "{att} {obj} is {att}" | **10.6** | **62.6** | 71.9 |

tributes. The proposed dataset is different from VAW in that it is 1) an open-vocabulary ranking task, instead of a fixed vocabulary domain classification task, and 2) has two variants, VGAR-A or VGAR-O focusing on either attribute recognition or object recognition given an attribute. This allows us to investigate how knowledge from pretraining differs between attribute concepts and object concepts. For VGAR-A, each ranking problem is formulated with respect to one object, with $N$ ground truth attributes taken from Visual Genome's annotations and $(50 - N)$ false attributes selected as those that are often associated with the given object but are not true for given image. VGAR-O mirrors this design, but is formulated with respect to an attribute present on the image. We obtain a dataset with 770,721 ranking problems for training, 7,997 for validation, and 32,299 for testing. Further details regarding dataset construction are provided in the supplementary materials, and we plan to make the preprocessed annotations publicly available upon acceptance.

## 4.2 RESULTS ON THE VAW DATASET

We show that generative prompting is better than contrastive prompting, then analyze various conditional models, and finally compare our results to the state-of-the-art.

**Generative v.s. Contrastive prompting.** The VAW dataset and the following metrics were used: rank (average rank of the correct predictions out of all 620 choices), mR@15 (mean recall over all classes at top 15 predictions for each instance), and mAP (mean average precision over all classes). We use rank as the primary metric as it is more comprehensive and better describes the ranking performance in a large candidate space.

Tab. 1 and 2 show the results of the zero-shot and fine-tuning settings, respectively. Generative prompting outperforms contrastive prompting in both settings, demonstrating a stronger ability to model fine-grained associations between objects and attributes. Generative prompting achieves a rank of 56.0 with its best prompt, compared to 95.1 (↓ lower is better) for contrastive prompting in the zero-shot setting (Tab. 1) and similarly achieves 10.6 vs 12.2 in the finetuning setting. (Tab. 2). As previously mentioned, there are two underlying reasons for generative prompting's superiority. First, it captures true visual attributes, while contrastive prompting may learn superficial connections through object identities (as shown in Tab. 1, adding object hints in contrastive prompting makes it perform worse). Second, it physically models the object-attribute relations through their dependencies and interactions, which eliminates counterfactual attribute-object pairs. Fig. 4 shows some qualitative examples of differences between generative prompting and contrastive prompting. In the top-left example, the contrastive prompting ranks highly the attributes "sky is bell shaped" and "sky is graffitied", which are highly associated with other objects present in the image but do not apply to the sky. This shows that contrastive prompting can surface attributes based on the associations acquired from contrastive pre-training, which is highly undesirable for attribute recognition.

**Conditional dependence modeling.** In Tab. 1 and 2, we also investigate the four types of graphical models (see Fig. 2) that generative prompting approximate. As finetuning results shows similar trends, we present the zero-shot results in Tab. 1 (bottom).

The simple classification prompt "{att}" does not model the important object prior and achieves the worst rank of 82.1. The PrefixLM prompt "{att} {obj}" produces a better model with a rank of 63.9, as it first classifies attributes then checks whether the attributes fit the "[MASK] {obj}". The MLM prompt "{obj} is {att}" has a similar rank of 61.9. We want to highlight that while

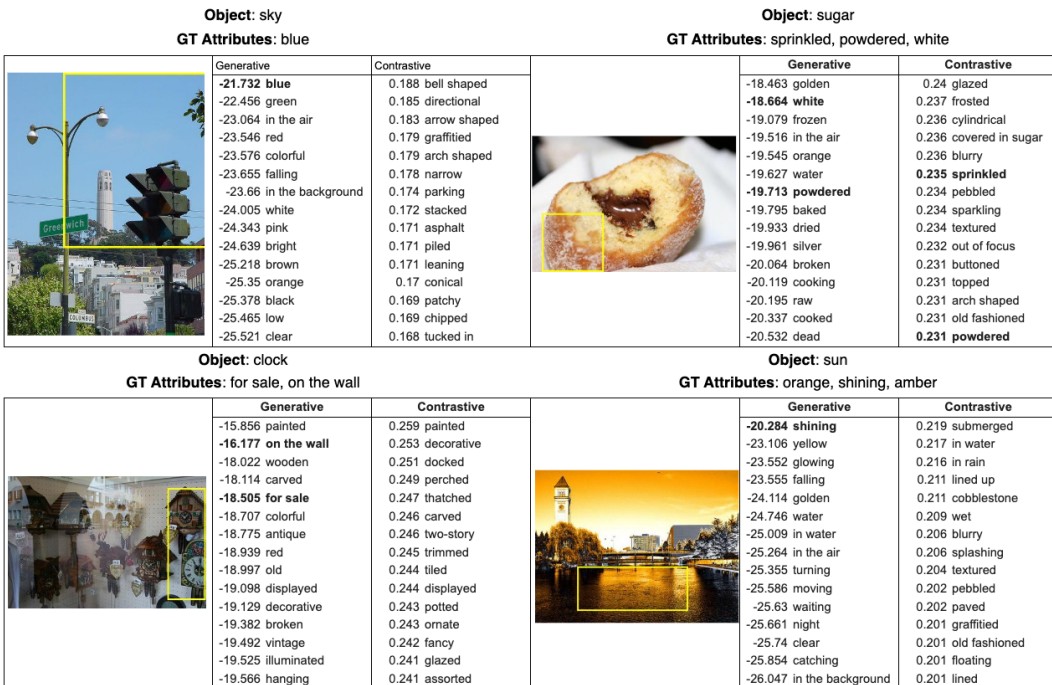

Figure 4: Zero-shot attribute prediction - qualitative results on the VAW dataset. Images are cropped using the yellow bounding boxes, and models only see the areas inside the boxes.

all baselines on the VAW in Tab. 3 are analogous to this formulation, it is not the best among the four graphical models. Therefore, improving the probability modeling in these SOTA methods can potentially further improve their performance, and our generative prompting offers an easy way to do so. Finally, the hybrid Prompt "{att} {obj} is {att}" performs the best with an average rank of 56.0. This is because it jointly considers three important factors: $p($ "{att}"$|v)$, $p($ "{obj}"$|v,$ "{att}"$)$, and $p($ "{att}"$|v,$ "{obj}"$)$, all captured by the proposed generative prompting. In particular, it is difficult for MLM to capture the latter two factors simultaneously, and this shows how our proposed prefixLM + generative prompting is a more flexible approach.

**Comparing to the SOTA.** We compared our fine-tuned model to the state-of-the-art methods in Tab. 3 using the following metrics from Pham et al. (2021): mAP (mean average precision over all classes), mR@15 (mean recall over all classes at top 15 predictions in each instance), mA (mean balanced accuracy over all classes), and F1@15 (overall F1 at top 15 predictions). The following baselines were considered: ResNet-Bas.-CE (Anderson et al., 2018; Jiang et al., 2020), ResNet-Bas. (Patterson & Hays, 2016), LSEP (Li et al., 2017), Sarafianos et al. (2018), PartialBCE + GNN (Durand et al., 2019), ML-GCN (Chen et al., 2019), SCoNE (Pham et al., 2021), and TAP (Pham et al., 2022). The authors from (Pham et al., 2021) are thanked for reimplementing/adapting all the baselines.

**Our model achieves the second place only slightly behind TAP, which was trained on an much larger and expensive fully-annotated attribute dataset.** Instead of TAP's elaborately designed object grounding, our method has a simple design that nevertheless incorporate more sophisticated probabilistic modeling. In addition, for the medium (72.0% mAP) and tail (60.6% mAP) attribute classes, our method shows significant improvements. This suggests a much stronger prior that allows the model to infer on rarely seen attributes.

### 4.3    RESULTS ON THE VGAR DATASET

**Generative vs Contrastive prompting.** We make similar observations as on the VAW dataset, shown in Tab. 4 and 5. We observe that generative prompts significantly outperformed the contrastive counterparts on both datasets. The best generative prompt on VGAR-A is "{att} {obj} is {att}", achieving a rank of 12.0, while the best one on VG Object Ranking is "{att} {obj}", achieving a rank of 5.8. This again verifies our claim that generative prompting is more optimal for attribute recognition than contrastive prompting.

Table 3: Comparing to the SOTA on the VAW dataset. The top rows show the baseline models; the last three rows shows the results of our method which finetunes the generative prompts. For mA, we report mA@threshold=0.005 as we cross-validated.

| Methods | Overall | | | | Class imb. mAP | | | Attribute types mAP | | | | | | |
|---|---|---|---|---|---|---|---|---|---|---|---|---|---|---|
| | mAP | mR$^{@15}$ | mA | F1$^{@15}$ | Head | Med. | Tail | Colo. | Mate. | Shap. | Size | Text. | Acti. | Other |
| ResNet-Bas.-CE | 56.4 | 55.8 | 50.3 | 61.5 | 64.6 | 52.7 | 35.9 | 54.0 | 64.6 | 55.9 | 56.9 | 54.6 | 47.5 | 59.2 |
| LSEP | 61.0 | 50.7 | 67.1 | 62.3 | 69.1 | 57.3 | 40.9 | 56.1 | 67.1 | 63.1 | 61.4 | 58.7 | 50.7 | 64.9 |
| PartialBCE + GNN | 62.3 | 52.3 | 68.9 | 63.9 | 70.1 | 58.7 | 40.1 | 57.7 | 66.5 | 64.1 | 65.1 | 59.3 | 54.4 | 65.9 |
| ResNet-Bas. | 63.0 | 52.1 | 68.6 | 63.9 | 71.1 | 59.4 | 43.0 | 58.5 | 66.3 | 65.0 | 64.5 | 63.1 | 53.1 | 66.7 |
| ML-GCN | 63.0 | 52.8 | 69.5 | 64.1 | 70.8 | 59.8 | 42.7 | 59.1 | 64.7 | 65.2 | 64.2 | 62.8 | 54.7 | 66.5 |
| Sarafianos et al. (2018) | 64.6 | 51.1 | 68.3 | 64.6 | 72.5 | 61.5 | 42.9 | 62.9 | 68.8 | 64.9 | **65.7** | 62.3 | 56.6 | 67.4 |
| SCoNE | 68.3 | 58.3 | 71.5 | **70.3** | **76.5** | 64.8 | 48.0 | 70.4 | **75.6** | 68.3 | **69.4** | 68.4 | 60.7 | 69.5 |
| TAP | **73.4** | **63.3** | 73.5 | **71.1** | - | - | - | - | - | - | - | - | - | - |
| Ours "{att} {obj}" | 70.8 | 61.8 | 73.7 | 68.3 | 74.0 | 71.0 | 58.2 | 73.1 | 75.0 | 70.9 | 61.8 | 72.2 | 68.8 | 70.7 |
| Ours "{obj} is {att}" | 72.0 | 62.1 | **74.7** | 68.7 | 74.9 | **72.0** | **60.6** | 75.2 | **76.0** | **72.6** | 62.9 | **72.7** | **69.6** | **72.0** |
| Ours "{att} {obj} is {att}" | 71.9 | **62.6** | 74.4 | 68.7 | **75.0** | **72.1** | 59.4 | **75.7** | 75.3 | 71.2 | 62.8 | 72.2 | **70.3** | 71.8 |

Table 4: Zero-shot prediction on VGAR-A.

| | Prompt Templates | Rank↓ | R$^{@1}$↑ | R$^{@5}$↑ | R$^{@10}$↑ |
|---|---|---|---|---|---|
| Cont. | "{att}" | 17.3 | 6.3 | 22.7 | 38.4 |
| | "{att} {obj}" | 16.1 | 9.7 | 28.9 | 43.6 |
| | "{obj} is {att}" | 17.2 | 8.7 | 26.4 | 40.7 |
| | "{att} {obj} is {att}" | 16.5 | 9.0 | 27.9 | 42.8 |
| Gene. | "{att}" | 14.0 | 8.9 | 34.2 | 53.0 |
| | "{att} {obj}" | **13.0** | 13.9 | 41.6 | **58.6** |
| | "{obj} is {att}" | 13.1 | **15.2** | **42.3** | **58.6** |
| | "{att} {obj} is {att}" | **12.0** | **17.6** | **46.6** | **62.2** |

Table 5: Zero-shot prediction on VGAR-O.

| | Prompt Templates | Rank↓ | R$^{@1}$↑ | R$^{@5}$↑ | R$^{@10}$↑ |
|---|---|---|---|---|---|
| Cont. | "{obj}" | 6.0 | 32.4 | 70.2 | 83.2 |
| | "{obj} is {att}" | 5.9 | 34.1 | 70.2 | 82.9 |
| | "{att} {obj}" | 5.9 | 35.3 | 70.9 | 83.2 |
| | "{att} {obj} is {att}" | 6.0 | 34.7 | 70.0 | 82.5 |
| Gene. | "{obj}" | 6.1 | 31.3 | 70.3 | 83.2 |
| | "{obj} is {att}" | **6.0** | 38.9 | **73.2** | **83.6** |
| | "{att} {obj}" | **5.8** | 40.6 | **74.2** | **84.4** |
| | "{att} {obj} is {att}" | 6.4 | **41.6** | 72.3 | 82.0 |

**Conditional dependence modeling.** Tab. 4 and 5 show the results on VGAR-A and VGAR-O. We boldface the targets to be predicted, which are *"{att}"* for VGAR-A, and *"{obj}"* for VGAR-O.

As in VAW, the simple classification prompt *"{att}"* or *"{obj}"* is the least effective, with a rank of 14.0 on VGAR-A and 6.1 on VGAR-O. The PrefixLM prompt *"{att} {obj}"* or *"{obj} is {att}"*, is significantly better as expected, with rank of 13.0 on VGAR-A and 6.0 on VGAR-O. This is achieved by first classifying the target token then checks whether the target token fits the context. However, the more optimally ordered MLM prompt *"{obj} is {att}"* or *"{att} {obj}"* outperforms the previous approach, which aligns with conclusion drawn by earlier works like Lampert et al. (2009); Jayaraman & Grauman (2014); Al-Halah et al. (2016) that suggest attributes help the classification of uncommon objects, hence our model's better-than-SOTA performance on the mid and long-tail categories in VAW. Notably, *"{att} {obj}"* achieves best performance on the VGAR-O. The Hybrid prompt *"{att} {obj} is {att}"* or *"{att} {obj} is {att}"* performs the best on VGAR-A with a rank of 12.0, but it falls behind the *"{att} {obj}"* variant on VGAR-O. We attribute this to the more challenging nature of attribute recognition as compared to object recognition, where the former can benefit from more complex dependency modeling introduced by our prompting, while the latter still relies more on salient information. The VGAR-A/VGAR-O experiments highlights the versatility of the proposed generative prompting, enabling us to predict attributes based on objects and vice versa. This flexibility demonstrates the all-encompassing, foundational nature of the prefixLM approach through generative prompting. By simply making changes to the prompt, we can construct various explainable probabilistic models, expanding the possibilities for modeling complex relationships between objects and attributes.

## 5 CONCLUSION

Our work propose to use prefixLM and a novel generative prompting mechanism for visual attribute recognition. By leveraging the complex word dependencies captured by prefixLM during pre-training, the proposed generative prompting enables the explicit modeling of various object-attribute dependencies in downstream attribute tasks. We showcase the flexibility of generative prompting in emulating various conditional dependencies, thereby unifying and simplifying the manually designed conditional dependencies. We envision that prefixLM + generative prompting can serve as a universal framework or meta-model for modeling complex logical relations.

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
