# ATTRIBUTE RECOGNITION WITH IMAGE-CONDITIONED PREFIX LANGUAGE MODELING

In this document, we include more information and details about our paper. We first discuss the broader impacts and the limitations. Then, we provide further experimental details and data preprocessing details of the proposed VGA dataset. We also provide additional qualitative experimental results and the license information of the images shown in the paper.

## 1 BROADER IMPACTS

Our work is dedicated to studying the visual attribute recognition problem in the context of large pretrained models. This carries potential social impact as advances in image-text large language modeling translates to stronger models that are better at capturing and encoding sensitive information. For example, in a text-only modeling context, Bard or ChatGPT oftentimes must process user inputs containing sensitive information such as race, politics, and personal identity. We are aware that our research focusing on extracting structural information from large language models can also be used to gather such sensitive information. Users could potentially use our research to gather and process information on the attributes of a person, a social group, or any visually or textually meaningful instances. It is critical that the research community is aware of these risks and community-driven supervision is present to avoid misuse.

## 2 LIMITATIONS

Our generative prompting approach is specifically designed for tasks where the assumed lengths of answers or prompts are similar. Since the sum of log probabilities in $L^{(gen)}$ is influenced by the length of the text, the approach is biased towards shorter answers. In the context of attribute prediction tasks, the assumption of similar lengths holds true, allowing us to treat attribute prompt optimization as joint probability optimization in a graph model. This task formulation sets it apart from VQA tasks, which typically involve multiple-choice questions with answers of varying lengths. It is worth noting that this limitation does not undermine our main contribution, which is the development of a novel adaptation framework that combines prefixLM pre-training with generative prompting for attribute recognition problems.

## 3 EXPERIMENTAL DETAILS

We finetune the model for the results shown in Tab. 2 and Tab. 3. The general methodology is described in Sec. 3.4 and the model architecture details are introduced in Sec. 4.1. We supplement the additional fine-tuning details in this section.

For the additional parameters $\mu_c$ and $\sigma_c$ in Sec. 3.4, we initialize $\mu_c$ using -15.0 and $\sigma_c$ using 0.5, inspired by the values we observed in Fig. 4 (which shows $-L^{(gen)}(v, t)$ for sorting purpose, will clarify in camera-ready). The initial values roughly transform the logits $p_c$ to zero mean and scale the standard deviation to 6.0. To finetune the model, we use a batch size of 4, a maximum text length of 16, a weight-decay of 0.01. We use the Adafactor optimizer with $\beta_1 = 0.9$ and $\beta_2 = 0.999$, and a learning rate of 1e-5 linearly decayed to zero in 100k training steps, which are roughly 1.8 training epochs. All experiments are conducted on single machine armed with TPUv3 with the average time to fine-tune a model being 7 hours.

## 4    DATA PREPROCESSING FOR THE VGA DATASET

To construct the two datasets, Visual Genome Attributes - attribute ranking (VGA-A) and Visual Genome Attributes - object ranking (VGA-O), we first extract object entities from each Visual Genome image annotation. For both datasets, we build the ground truth set of attribute-object pairings by coupling the entity name with each of the entity's list of attributes. We then expand this set of attribute-object pairings to make up a 50-choice ranking problem by adding to it fake pairings. For VGA-A, we pair the object name with attributes that are inapplicable in the context of the image, and for VGA-O, we pair the correct attributes to inapplicable object names. To make the problem challenging, these false pairings are selected in accordance to the dataset's conditional probability $P(object|attribute)$ or $P(attribute|object)$, i.e. for a given object or attribute, we select from top to bottom attribute-object pairings that are most likely to occur in the Visual Genome dataset. If a selected attribute-object pairing exists on the current image, either on the current object/attribute or on another object/attribute, we do not add this pairing to the set of 50 as a fake pairing. And if the given object or attribute does not appear often in Visual Genome and there is not enough fake pairings to make up the 50 choices, we then add objects or attributes according to the dataset prior $P(object)$ or $P(attribute)$ to fill the rest of the choices. For both VGA-A and VGA-O, we obtain 770,721 entities for training, 32,299 entities for testing, and 7,997 entities for validation. The splits and the generated annotations for VGA will be made publicly available to help others to reproduce our experiments and to conduct further research on this challenging dataset.

## 5    MORE QUALITATIVE EXAMPLES

We provide more examples to compare our zero-shot prompting methods, we also include the results from the fully-supervised method SCoNE [14] trained on the VAW dataset. Fig. 1 shows the results. Some interesting observations can be made. First, VAW is still a closed domain dataset, lacking in the coverage of long-tailed attributes. In example (2), our generative prompting predicts "decorative", "antique", and "bamboo", which are visually salient and grammatically correct. However, the ground-truth annotation does not include these two options. Second, compared to others, generative prompting can surface some of the most significant attributes in the examples. For example, "in the background", "decorative", "worn", or "closed". However, many predictions of the contrastive prompting method are visually imperceptible or incorrect, such as arch-shaped, standing, partially-eaten, water.

| | Generative | Contrastive | SCoNE[14] |
|---|---|---|---|
| **(1) Object: mountain** GT Attributes: tree-covered | -21.891 in the background | 0.197 arch shaped | **0.998 tree-covered** |
| | -23.587 green | **0.197 tree-covered** | 0.994 green |
| | -23.649 for sale | 0.196 stucco | 0.989 grassy |
| | -23.744 blue | 0.196 red striped | 0.972 in the background |
| | -24.557 water | 0.194 cylindrical | 0.963 full |
| | -24.598 small | 0.193 partially visible | 0.943 far away |
| | -24.677 red | 0.193 trimmed | 0.901 wide |
| | -24.993 relaxing | 0.193 side view | 0.893 tall |
| | -25.057 white | 0.191 statue | 0.688 lush |
| | -25.085 closed | 0.191 displayed | 0.655 dense |
| | -25.109 orange | 0.191 graffitied | 0.631 large |
| | -25.17 open | 0.19 looking down | 0.587 dark |
| | -25.304 clear | 0.189 looking up | 0.575 rocky |
| | -25.328 in the air | 0.189 rolled up | 0.434 leafy |
| | -25.396 sleeping | 0.187 wallpapered | 0.427 small |
| **(2) Object: lamp** GT Attributes: vertical, amber, orange | -19.623 decorative | 0.252 wicker | 0.982 standing |
| | -19.842 bronze | 0.251 trimmed | **0.979 orange** |
| | -19.907 antique | 0.249 displayed | 0.973 bright |
| | -19.908 for sale | 0.246 tucked in | 0.959 illuminated |
| | -20.316 used | 0.245 wallpapered | 0.946 golden |
| | -20.316 white | 0.244 decorative | 0.935 shaded |
| | -20.322 wooden | 0.244 wispy | 0.93 modern |
| | -20.338 golden | 0.243 cushioned | 0.921 thin |
| | -20.452 small | 0.243 resting | 0.921 yellow |
| | -20.576 painted | 0.242 pinned | **0.901 vertical** |
| | -20.606 open | 0.242 pinstriped | 0.871 small |
| | -20.715 bamboo | 0.241 upholstered | 0.778 brown |
| | -20.717 yellow | 0.241 buttoned | 0.771 rounded |
| | -20.771 on the wall | 0.241 unlit | 0.701 tall |
| | -20.811 hanging | 0.241 bamboo | 0.667 tiny |
| **(3) Object: tail** GT Attributes: patterned, spotted, hanging | -22.257 striped | 0.276 striped | 0.992 hairy |
| | **-22.457 spotted** | 0.257 blue striped | **0.969 hanging** |
| | -22.583 upside down | 0.257 barred | 0.951 long |
| | -22.66 brown | 0.257 red striped | 0.915 small |
| | -22.994 running | **0.25 spotted** | 0.907 black |
| | -23.208 jumping | 0.242 partially eaten | 0.898 extended |
| | -23.252 flying | 0.241 male | 0.897 dark colored |
| | -23.628 walking | 0.24 resting | 0.896 bushy |
| | -23.692 falling | 0.24 lined up | 0.896 dark |
| | -23.692 broken | 0.24 camouflage | 0.878 fluffy |
| | -23.752 white | 0.239 hiding | 0.803 brown |
| | -23.852 open | 0.239 pinstriped | **0.775 patterned** |
| | -24.181 black | 0.238 slender | 0.768 gray |
| | -24.192 dead | 0.238 piled | 0.621 curved |
| | -24.303 painted | 0.238 horned | 0.595 fuzzy |
| **(4) Object: shoes** GT Attributes: athletic | -24.524 black | 0.204 skateboarding | 0.998 black |
| | -24.528 broken | 0.201 circular | 0.987 raised |
| | -24.53 worn | 0.2 cylindrical | 0.949 used |
| | -24.707 leather | 0.198 bell shaped | 0.926 dark |
| | -24.89 in the air | 0.198 bending | 0.844 worn |
| | -24.936 dead | 0.197 knocked over | **0.749 athletic** |
| | -24.998 cut | 0.197 bent | 0.709 leather |
| | -25.037 white | 0.196 pulled back | 0.669 trimmed |
| | -25.515 old | 0.195 pinned | 0.666 dark colored |
| | -25.597 used | 0.195 holed | 0.579 gray |
| | -25.7 flying | 0.194 operating | 0.567 close |
| | -25.738 falling | 0.193 cooked | 0.472 shiny |
| | -25.881 painted | 0.193 skating | 0.456 brown |
| | -25.894 flat | 0.192 cutting | 0.421 old |
| | -26.036 vintage | 0.192 stopped | 0.415 wet |
| **(5) Object: hydrant** GT Attributes: tall, clean, close, thin, red, painted, metal, yellow, hard | **-15.886 red** | 0.302 displayed | 0.909 water |
| | -17.218 closed | 0.3 light skinned | 0.858 buried |
| | -17.521 broken | 0.299 tagged | **0.857 metal** |
| | **-17.627 painted** | 0.297 vertical | 0.838 colorful |
| | -17.72 for sale | 0.297 pinstriped | **0.83 tall** |
| | -17.77 empty | 0.296 lined | 0.83 old |
| | -17.992 open | 0.296 lined up | **0.814 red** |
| | -18.023 old | 0.295 modern | **0.806 painted** |
| | -18.031 orange | 0.295 docked | **0.766 thin** |
| | -18.285 water | 0.295 neat | 0.686 standing |
| | -18.309 dead | 0.295 amber | 0.627 large |
| | -18.377 mounted | **0.295 painted** | 0.617 shiny |
| | -18.509 upside down | 0.295 old fashioned | 0.602 bright |
| | -18.703 funny | 0.295 full | 0.492 tagged |
| | -18.723 in the background | **0.295 tall** | 0.465 dirty |

Figure 1: More qualitative examples on the VAW dataset. **Generative** and **Contrastive** use zero-shot prompting while baseline SCoNE [14] is trained on the VAW dataset.

## 6 IMAGE ATTRIBUTION

In this paper we display several images from the VAW dataset. The Flickr links and the license information for these images can be found in Tab. 1. We thank the original photographers for sharing their photos.

Table 1: Flickr links and license of the images.

| Flickr link | User | License |
|---|---|---|
| **Paper Fig. 4 (from left to right, top to bottom)** | | |
| https://www.flickr.com/photos/mount_otz/31929683/ | mount_otz | CC BY-NC-SA 2.0 |
| https://www.flickr.com/photos/jenny-pics/2381135314/ | jenny-pics | CC BY 2.0 |
| https://www.flickr.com/photos/worldofjan/2984166899/ | worldofjan | CC BY-NC 2.0 |
| https://www.flickr.com/photos/23909838@N02/3363471858/ | 23909838@N02 | CC BY-SA 2.0 |
| **Supplementary materials Fig. 1 (from top to bottom)** | | |
| https://www.flickr.com/photos/felipelopez/2660779383/ | felipelopez | CC BY-NC 2.0 |
| https://www.flickr.com/photos/afagen/2269170288/ | afagen | CC BY-NC-SA 2.0 |
| https://www.flickr.com/photos/nbarcet/2172355975/ | nbarcet | CC BY 2.0 |
| https://www.flickr.com/photos/dammit_jack/1523816737/ | dammit_jack | CC BY-NC 2.0 |
| https://www.flickr.com/photos/mjhagen/4347200481/ | mjhagen | CC BY 2.0 |

## REFERENCES