# OpenReview forum: "Attribute Recognition with Image-Conditioned Prefix Language Modeling"
_ICLR.cc/2024/Conference — Submitted to ICLR 2024_

### Official Review · Reviewer_pzC1 · 2023-10-24

**Soundness:** 3 good
**Presentation:** 2 fair
**Contribution:** 2 fair
**Rating:** 5
**Confidence:** 4

**Summary:**

This paper proposed an image captioning way to address object attribute recognition and object-attribute compositional learning. Given the limitations of previous methods like the CLIP-based method, the authors proposed a generative prompting method to solve both the classification and compositional learning problems. On two datasets, the proposed method was compared with recent works and showed improvements.

**Strengths:**

+ Given the current development in LLM, using the captioning solution to address the visual understanding is non-trivial. This work used a general pipeline to address a classical visual relation and recognition problem.

+ The discussion about the CLIP-based works makes sense.

**Weaknesses:**

- Though the solution is sound according to the new development tendency, I do not find too many new insights and "surprising" parts. There are many works using captioning via LLM to solve nearly all other directions in visual understanding, e.g., action/object recognition, visual relationship understanding, VQA, etc. Please give a discussion covering more domains to analyze the contribution.

- Compositional learning and its zero-shot setting (CZSL) is challenging with previous paradigms given the fixed train and test sets. Now, we have huge datasets like LAION and many other datasets beyond images like text. More discussion or insights about the new weapons and the CZSL would be a better contribution.

- Though previous non-LLM works are "old", they can still be used as the baselines.

- The method part is kinda of too brief.

- More experiments on more datasets would be more solid to support the claims.

- typo: many wrong quotation marks ”xx”.

- some weird green boxes appeared.

**Questions:**

- Using huge LAION, then the fairness consideration in the experiments?

- Detailed analysis of the long-tailed distribution and results?

---

> ### Author Response · Authors · 2023-11-23
>
> We want to first thank the reviewers for recognizing our work’s novel **reformulation of the attribute learning problem** and its contribution in successfully **leveraging pretrained large language models for fine-grained visual understanding** through generative prompting. We also thank the reviewers for recognizing the significance of our experimental results, which 1) shows strong performance on par with SOTA, **despite not having task-specific modules or pretraining with annotated in-domain datasets**, and 2) supports our claim that CLIP representation alone is ill-suited for attribute learning and that **our proposed generative prompting can bridge the gap in performance**. And lastly, we thank the reviewers for acknowledging our benchmark’s broad usefulness to the community, as it unifies object and attribute recognition into a single setting and allows for direct comparisons that can measure the level of disentanglement between attribute and object representations.
>
> \
> **Q1:** Thank you for the comment. We want to clarify that the main insight of our work is the formulation of attribute learning as a knowledge retrieval problem to extract relevant knowledge embedded in large vision-language models (VLM) through the means of generative prompting. We also establish the **fundamental connection between conditional probability graphs and prompt template construction** in the context of prefix language models, which we illustrate in figure 2, and recognize its importance and applications for visual attribute learning. To further elaborate, this formulation allows us to flexibly extract relevant knowledge of object-attribute-image relationships from the pretrained foundational models for downstream tasks outside of the foundation model settings, as we mentioned in paragraph 4 of the introduction. To our knowledge, we are the first to propose this **novel view of treating attribute learning as a language modeling and knowledge retrieval problem** and in applying VLMs towards visual attribute recognition, as we discussed in paragraph 4 of Related Work.
>
> To further highlight the difference between our work and other captioning work in visual understanding, we would like to reiterate parts of the Approach section in the paper. One of our central ideas is to formally express attribute learning as a task of learning image-object-attribute conditional probabilities, which naturally leads to generative prompting in a learning distillation paradigm  - constructing custom probability dependencies between objects and attributes at inference time (with 4 examples given in section 3.3 and in figure 2) and use it to extract relevant knowledge on image-object-attribute relationship acquired during standard large-scale pretraining in the foundation VLM model, as we discussed in paragraph 6 in section 3.3 and illustrated in figure 1.
>
> \
> **Q2:** Thank you for the comment. We will include additional discussions and citations on compositional zero-shot learning in the final submission and discuss how it relates to attribute learning and our work.
>
> \
> **Q3:** We have included numerous non-LLM baselines in our experiment table for comparison. We would love to add additional important ones we have not yet included if you could kindly bring them to our attention.
>
> \
> **Q4:** We will expand and elaborate the method section in our final submission.
>
> \
> **Q5:** We examined a list of potential dataset and found HICO and V-COCO to be more relevant, but not necessary for expanding on for our work’s central idea due to overlapping domain (COCO) and the focus on relational attributes.
>
> \
> **Q6 Q7:** We will fix these issues in the final submission.
>
> \
> **Q8:** LAION is a widely-accepted and widely-used dataset for pretraining, so our method would be considered as standard and fair.
>
> \
> **Q9:** Thank you for the comment. Our model performs significantly better on the less frequent categories in the distribution long tail, as shown in table 3. SCoNE and TAP’s slightly higher overall mAP and head mAP does not necessarily mean they are superior models, since they represent performance on frequently observed categories where it is easy for models to **fit to the dataset**. Our model’s performance on the long tail shows that it fits more to the prior knowledge carried by the foundation models.
>
> Please see below for qualitative results on the least frequent attributes in VAW:
> | Attribute   | SCoNE mAP | Our mAP |
> |-------------|-----------|---------|
> | nylon       |    0.6984 |  0.5333 |
> | bell shaped |    0.6955 |  0.9167 |
> | braided     |    0.3893 |  0.7046 |
> | styrofoam   |    0.3591 |  0.3354 |
> | spiral      |    0.2294 |  0.8605 |
> | kissing     |    0.0409 |  0.4085 |
> | wallpapered |    0.5293 |  0.8956 |
> | smoking     |    0.1966 |  0.3671 |
> | stucco      |    0.3774 |  0.5914 |
> | cubed       |    0.1102 |  0.4258 |
> | **TAIL MEAN**   |     **0.480** |   **0.594** |

---

> > ### Comment · Reviewer_pzC1 · 2023-11-23
> > **Post-rebuttal**
> >
> > Thanks for the response. After reading the reviews and responses, I tend to keep my original rating.

---

### Official Review · Reviewer_fA8W · 2023-10-31

**Soundness:** 2 fair
**Presentation:** 3 good
**Contribution:** 2 fair
**Rating:** 5
**Confidence:** 4

**Summary:**

While zero-shot object recognition has largely been solved by large language-vision models such as CLIP, visual attribute recognition remains challenging because CLIP’s contrastively learned representations do not effectively encode object-attribute dependencies.
In this paper, the authors revisit the problem of attribute classification and propose a solution using generative prompting, which revolves around a strategy for measuring the probability of generating prompts.
Unlike contrastive prompting, generative prompting is order-sensitive, and its design reflects the downstream requirements of object-attribute decomposition.
The authors demonstrate through experiments that generative prompting outperforms contrastive prompting on two datasets that require visual reasoning, Visual Attribute in the Wild (VAW), and a modified formulation of Visual Genome (VGAR).

**Strengths:**

- The proposed method is easy to understand.
- In terms of attribute recognition research. The proposed approach with generative prompting seems new.

**Weaknesses:**

- As a machine learning research, the technical significance and novelty seem weak. This work looks like a simple prompt engineering paper to me. To claim the technical significance, the authors should try with more various prompts and compare the results.

- Also, as the main contribution of this paper is to replace contrastive prompting with generative prompting for attribute recognition, the authors should provide a theoretical explanation of why generative prompting is better than contrastive prompting. Otherwise, the technical contribution might be weak as a machine learning paper.

- The experiment is also weak. In Table 3, the performance of the proposed method is not noticeably better than the baseline models (e.g., TAP).

- Finally, there should be more comprehensive experimental results. For example, how about the result with other LLM models than CoCa? Also, in Tables 4 and 5, why didn’t the authors compare the proposed model with the state-of-the-art methods? Finally, it would be better to evaluate the proposed method on other attribute recognition datasets, such as UTZappos or MIT datasets, or HOI detection datasets, such as HICO or V-COCO datasets.

**Questions:**

Please refer to the questions in the weakness.

---

> ### Author Response · Authors · 2023-11-23
>
> We want to first thank the reviewers for recognizing our work’s novel **reformulation of the attribute learning problem** and its contribution in successfully **leveraging pretrained large language models for fine-grained visual understanding** through generative prompting.. We also thank the reviewers for recognizing the significance of our experimental results, which 1) shows strong performance on par with SOTA, **despite not having task-specific modules or pretraining with annotated in-domain datasets**, and 2) supports our claim that CLIP representation alone is ill-suited for attribute learning and that **our proposed generative prompting can bridge the gap in performance**. And lastly, we thank the reviewers for acknowledging our benchmark’s broad usefulness to the community, as it unifies object and attribute recognition into a single setting and allows for direct comparisons that can measure the level of disentanglement between attribute and object representations.
>
> \
> **Q1:** Thank you for your comment. We want to clarify that the main contribution of our work is not the prompt engineering, but the formulation of attribute learning as a knowledge retrieval problem to extract relevant knowledge embedded in large vision-language models (VLM) through the means of generative prompting. Our work advances the state of the art by reframing the attribute learning problem to **solve it through leveraging standard pretrained VLMs**. We also establish the **fundamental connection between conditional probability graphs and prompt template construction** in the context of prefix language models, which we illustrate in figure 2, and recognize its importance and applications for visual attribute learning. To further elaborate, this formulation allows us to flexibly extract relevant knowledge of object-attribute-image relationships from the pretrained foundational models for downstream tasks outside of the foundation model settings, as we mentioned in paragraph 4 of the introduction. To our knowledge, we are the first to propose this **novel view of treating attribute learning as a language modeling and knowledge retrieval problem** and in applying VLMs towards visual attribute recognition, as we discussed in paragraph 4 of Related Work.
>
> \
> **Q2:** Thank you for your comment. To reiterate section 3.1 on page 4, generative prompting is better than contrastive prompting because it **explicitly decomposes and represents the dependency in text input** by taking word order and relation into consideration, which also **align closer with a prefixLM’s pretraining-acquired knowledge** of diverse compositions of object-attribute dependencies. Given a nonsensical pair like “sky is parking”, the conditional probability term for “parking”, P(“parking” | v, s0, “sky”, “is”), would make the joint probability very small. CLIP-based contrastive prompting does not model dependencies in the text and has the problem of discrepancy between pretraining objective and downstream tasks, as it learned to align text to image without structures, resulting in a focus on objects, but is later tasked with understanding finer attributes that were not salient in pretraining objectives.
>
> \
> **Q3:** Thank you for the comment. Our model performs significantly better on the less frequent categories in the distribution long tail, as shown in table 3. SCoNE and TAP’s slightly higher overall mAP and head mAP does not necessarily mean they are superior models, since they represent performance on frequently observed categories where it is easy for models to **fit to the dataset**. Our model’s performance on the long tail shows that it fits more to the prior knowledge carried by the foundation models.
>
> Additionally, the methods should not be directly compared because our focuses on **cross-domain knowledge extraction and learning distillation**, instead of constructing a task-specific model with specialized modules and training procedures. SCoNE and TAP both rely on object mask supervision and custom datasets during training, require task-specific modules, and are trained and tested in the same domain.
>
> \
> **Q4:** Thank you for the comment. CoCa was selected because it integrates both CLIP based training as well as captioning based training in the same model, allowing for a fair comparison, and because we can finetune it for comparisons with baselines.
>
> Table 4 and 5 highlights the difference between probability metamodels in a zero-shot setting. Most of the baselines do not have zero-shot capability. TAP does (embedding space projection, similar to contrastive prompting), but its implementation is not available to us.
>
> Thank you for the list of potential datasets we can explore. We carefully examined each dataset and found HICO and V-COCO to be more relevant, but not necessary for expanding on for our work’s central idea due to overlapping domain (COCO) and the focus on relational attributes.

---

### Official Review · Reviewer_yHgE · 2023-11-01

**Soundness:** 3 good
**Presentation:** 2 fair
**Contribution:** 3 good
**Rating:** 8
**Confidence:** 3

**Summary:**

While large vision-language models like CLIP excel at zero-shot object recognition, they struggle with zero-shot visual attribute recognition due to an inability to effectively encode object-attribute relationships. This paper tackles this challenge by introducing generative prompting. This approach redefines attribute recognition by assessing the likelihood of generating prompts that express the attribute relation. Unlike its counterpart, contrastive prompting, generative prompting is order-sensitive and tailored for object-attribute decomposition. Experimental results reveal that generative prompting surpasses contrastive prompting in performance on two visual reasoning datasets: Visual Attribute in the Wild (VAW) and a newly introduced version of Visual Genome, termed Visual Genome Attribute Ranking (VGAR).
The paper also shows strong performance against SOTA, despite being trained without annotated data.

**Strengths:**

1. The paper also shows strong performance against SOTA, despite being trained without annotated data.
2. The paper has clear ablations to show the value of their proposed method
3. In general, the key idea of using generative prompting/modeling to solve complex localized reasoning tasks is an interesting direction
4. The VGAR benchmark which unifies object and attribute recognitions, is broadly useful for the community

**Weaknesses:**

I don't see any major weaknesses in the paper

**Questions:**

N/A

---

> ### Author Response · Authors · 2023-11-23
>
> We want to thank the reviewer for recognizing our work’s novel reformulation of the attribute learning problem and its contribution in successfully leveraging pretrained large language models for fine-grained visual understanding through generative prompting. We agree that the paper’s key idea of using generative prompting/modeling to solve complex localized reasoning tasks is an interesting and promising direction of exploration for the community to explore. We also thank the reviewers for recognizing the significance of our experimental results, which 1) shows strong performance on par with SOTA, despite not having task-specific modules or pretraining with annotated in-domain datasets, and 2) supports our claim that CLIP representation alone is ill-suited for attribute learning and that our proposed generative prompting can bridge the gap in performance. And lastly, we thank the reviewers for acknowledging our benchmark’s broad usefulness to the community, as it unifies object and attribute recognition into a single setting and allows for direct comparisons that can measure the level of disentanglement between attribute and object representations.

---

### Official Review · Reviewer_ZMSV · 2023-11-06

**Soundness:** 2 fair
**Presentation:** 3 good
**Contribution:** 2 fair
**Rating:** 5
**Confidence:** 4

**Summary:**

This work is about rcognizing image attributes through utilizing language models. Some experiments are shown for the results, and compared with other published works.

**Strengths:**

The use of existing language models for recognizing image attributes is interesting.

The experimental comparisons are important.

**Weaknesses:**

In my understanding, the main contribution of the work is the prompt engineering, i.e., designing prompt for accessing large language models, which is interesting and useful, but it is not develping a new mothod to advance the state-of-the-art, from the algorithm or theory viewpoint. Thus if my understanding is correct, the contribution of the paper is not at the level of ICLR.

To my knowledge, there are already some existing companies that work on prompt engineering for a better use of the ChatGPT or other large language models. Thus the prompt engineering is not that new, even from the application point of view.

It could be better if the prompting engineering shown in the work can be combined with some novel algorithm development, the paper might be a better contribution to the ICLR conference.

**Questions:**

As my concerns shown in the Weakness part.

---

> ### Author Response · Authors · 2023-11-23
>
> We want to first thank the reviewers for recognizing our work’s novel **reformulation of the attribute learning problem** and its contribution in successfully **leveraging pretrained large language models for fine-grained visual understanding** through generative prompting. We agree that the paper’s key idea of using generative prompting/modeling to solve complex localized reasoning tasks is a promising direction of exploration. We also thank the reviewers for recognizing the significance of our experimental results, which 1) shows strong performance on par with SOTA, **despite not having task-specific modules or pretraining with annotated in-domain datasets**, and 2) supports our claim that CLIP representation alone is ill-suited for attribute learning and that **our proposed generative prompting can bridge the gap in performance**. And lastly, we thank the reviewers for acknowledging our benchmark’s broad usefulness to the community, as it unifies object and attribute recognition into a single setting and allows for direct comparisons that can measure the level of disentanglement between attribute and object representations.
>
> \
> **Q1:** Thank you for your comment. We want to clarify that the main contribution of our work is not the prompt engineering, but the formulation of attribute learning as a knowledge retrieval problem to extract relevant knowledge embedded in large vision-language models (VLM) through the means of generative prompting. Our work advances the state of the art by reframing the attribute learning problem to **solve it through leveraging standard pretrained VLMs**. We also establish the **fundamental connection between conditional probability graphs and prompt template construction** in the context of prefix language models, which we illustrate in figure 2, and recognize its importance and applications for visual attribute learning. To further elaborate, this formulation allows us to flexibly extract relevant knowledge of object-attribute-image relationships from the pretrained foundational models for downstream tasks outside of the foundation model settings, as we mentioned in paragraph 4 of the introduction. To our knowledge, we are the first to propose this **novel view of treating attribute learning as a language modeling and knowledge retrieval problem** and in applying VLMs towards visual attribute recognition, as we discussed in paragraph 4 of Related Work.
>
> To further address your concern on whether our work develops a new method or advances the state of the art, we would like to reiterate parts of the Approach section in the paper. We formally express attribute learning as a task of learning image-object-attribute conditional probabilities, which can be solved in a novel pretraining-to-distillation paradigm with our proposed method. Our method constructs custom probability dependencies between objects and attributes at inference time (with 4 examples given in section 3.3 and in figure 2) and use it to extract relevant knowledge on image-object-attribute relationship acquired during standard large-scale pretraining in the foundation VLM model, as we discussed in paragraph 6 in section 3.3 and illustrated in figure 1. This **theoretical and practical reframing of the attribute learning problem** and the proposed method of applying and **extracting knowledge from LLMs** towards the task through generative prompting are both novel and contribute to advancing the state of the art.
>
> \
> **Q2:** Thank you for your comment. We invested a large amount of time but did not find any public, related work on prompt engineering in the context of extracting knowledge from vision-language models to improve attribute learning problems. To clarify, our work is not a paper on prompt engineering, since its central idea is to introduce a new way of thinking about the attribute learning problem **through a probabilistic modeling perspective**, as we first discussed in paragraph 4 of Related Work and expanded upon in paragraph 6 of section 3.3 in Approach. As part of the demonstration of our formulation and method’s capabilities, we study several possible conditional probability graphs at inference time, expressed through different constructions of the generative prompt. We conduct thorough experiments in section 4.2 and 4.3 to show how model performance can be improved by just improving how knowledge is extracted from the foundation model, namely by studying and improving the conditional metamodel we use to retrieve information. Lastly, we also want to mention that besides zero-shot experiments, our work also dedicated half of section 4 on fine-tuning experiments, which goes beyond what prompt engineering papers study.
>
> \
> **Q3:** Thank you for the comment. As we also discussed above, the prompt study itself is not the focus. Our main contribution remains to be the reframing of the attribute learning problem and the proposed method of extracting and applying LLM knowledge towards the task.

---

### Meta-Review · Area_Chair_BAQ8 · 2023-12-06

**Metareview:**

This paper presents a method for attribute recognition with image-conditioned prefix language modeling.  The paper was reviewed by four reviewers, who gave scores of 5, 8, 5, 5.  Strengths mentioned by the reviewers include the promising direction of using generative prompting for zero-shot visual attribute recognition, and some good results.  Weaknesses mentioned by the reviewers include limited technical contribution, weak experimental evaluation, missing experiments, and limited insights.  The rebuttal, and the authors' message to the ACs, partly addressed the issue of missing experiments, and to some degree the limited contribution and insights.  However, limited technical contribution was not fully addressed, and neither was the weak experimental validation -- both of these points were mentioned by reviewers in their final post-rebuttal discussion.  The reviewer who gave the sole acceptance score of 8 had limited confidence (3).  The paper, rebuttal, discussion, and author messages were carefully discussed among the ACs, and the ACs agree that the paper has limited novelty and the experimental validation are not strong enough to support acceptance. The ACs also recommend that the authors try to reposition the paper so that the main technical contributions are more directly understandable by the readers as intended by the authors.  The ACs would like to encourage the authors to improve the paper and resubmit to another conference.

**Justification For Why Not Higher Score:**

The paper is not ready for acceptance based on the weaknesses mentioned above.

**Justification For Why Not Lower Score:**

N/A

---

### Decision · Program_Chairs · 2024-01-16

Reject